# Development of the Virtual Reality Application: "The Ships of Navarino"

Orestis Liaskos [1,*], Sofia Mitsigkola [1], Andreas Arapakopoulos [2], Georgios Papatzanakis [1], Alexandros Ginnis [1], Christos Papadopoulos [1], Sofia Peppa [2] and Georgios Remoundos [3]

[1] School of Naval Architecture and Marine Engineering, National Technical University of Athens, 10682 Athens, Greece; smitsigkola@gmail.com (S.M.); gpap@deslab.ntua.gr (G.P.); ginnis@naval.ntua.gr (A.G.); chpap@central.ntua.gr (C.P.)

[2] Department of Naval Architecture, University of West Attica, 12243 Athens, Greece; aarapakopoulos@uniwa.gr (A.A.); speppa@uniwa.gr (S.P.)

[3] Department of Shipping, Trade and Transport, School of Business, University of the Aegean, 82132 Chios, Greece; gremoundos@aegean.gr

* Correspondence: liaskos.orestis@gmail.com; Tel.: +30-697-848-3826

**Abstract:** Virtual reality and 3D modeling techniques are increasingly popular modes of representation for historical artifacts and cultural heritage, as they allow for a more immersive experience. This article describes the process that was adopted for the development of a virtual reality application for four ships involved in the historic battle of Navarino. The specific naval battle was the culmination of military operations during the Greek Revolution in 1827, in which the allied British, Russian, and French fleet defeated Turkish-Egyptian forces. Representative 3D models of four significant warships that participated in the battle of Navarino were created: the British "Asia", the French frigate "Armide", the Russian "Azov", and the Ottoman "Kuh-I-Revan". These historic ships were digitally designed according to historical drawings and a VR battle environment was developed, which visitors can experience. In addition, the 3D models were generated by a 3D printer and painted according to the digitized ship-models. The development was conducted within the realm of the NAVS Project. The VR application, "The Ships of Navarino", as well as the 3D-printed models were presented as part of a physical exhibition hosted in the Eugenides Foundation in Athens, Greece.

**Keywords:** virtual reality; cultural heritage; 3D modeling; game engines; application development; ship design; 3D printing; exhibition

## 1. Introduction

Positioned in the intersection of technological and other sectors, VR applications are often in the middle of multidisciplinary stakeholders, fulfilling various needs and addressing complicated issues. The VR experience "The Ships of Navarino" was developed for the Oculus Rift S and the Oculus Quest, to be presented as a part of a physical exhibition held in the Eugenides Foundation, Athens, Greece. More specifically, the VR experience "The Ships of Navarino" is a tour inside four significant ships that participated in the battle of Navarino: the British "Asia", with admiral Cordington in the lead, the French frigate "Armide", the Russian "Azov", and the Ottoman "Kuh-I-Revan". Considering that the VR experience was a part of a larger exhibition, it should have a limited duration and be accessible to all audiences.

It is difficult to visualize a historical artifact when there are technological limitations. Optimization processes will often boost the performance based on the cost of visual fidelity and the risk of historical inaccuracies. This paper will highlight the development process of the recent VR project "The Ships of Navarino" and will provide an empirical contribution to the optimization methodology of a VR application that communicates intact historical information and opens up the possibilities of a broader dissemination range. During this

project, based on historical sources, four ships involved in the historic Battle of Navarino were digitally recreated and developed into a VR application that became accessible to a wide audience as part of the anniversary exhibition "Run onto the waves of the formidable sea. 1821, the War at Sea." in the Eugenides Foundation [1]. The optimization methodologies adopted targeted the VR platforms and resulted in 3D models also suitable for online hosting, enriching the dissemination of the project. Furthermore, based on the 3D models and 3D textures, accurate models of the ships were 3D-printed and painted for display in the exhibition.

## 1.1. Related Works

Virtual Reality (VR), 3D modeling techniques, and 3D printing have become very popular in various fields such as cultural heritage [2–6], ship design [7–9], and education [10–12]. Museums and institutions [13] use new media to offer new ways to communicate historical information to visitors and engage with new audiences. Information and communication technologies (ICTs) are known to enhance creativity in cultural and educational experiences, as they encourage learning by exploring an open environment and allow for a nonlinear storyline [14]. The use of VR enhances the user's immersion, while 3D models hosted online broaden the accessibility of information to users situated outside the reach of a cultural institution [15] (p. 7002). Furthermore, historical artifacts can be replicated and restored by using 3D printing technology [16].

## 1.2. The Ships of Navarino

Requirements and constraints, such as the time duration of the VR visit, the inclusion of people not accustomed to watching VR, and the device specifications set some early design directions for the general approach.

Specifically, considering the time limitation, unnecessary interaction with the VR environment or any gamification elements that would extend the duration of the guide were not included in the final experience. Design decisions were aimed towards a more friendly approach for people new to VR, such as the transition between ships. The VR hands were included, providing a better understanding of the controller locations and to facilitate actions such as pointing. Moreover, the introduction scene was designed to resemble a cabin (Figure 1), as this is the only ship room that is, scale-wise, closer to the human size, offering a more friendly transition from the real world to the virtual one.

Although interacting with the virtual world provides many possibilities for viewers to experience historical events and artifacts, the available technology comes with significant limitations regarding performance. While the Oculus Rift S targets 1–2 million triangles and 500–1000 draw calls per frame, the Oculus Quest targets 300,000–500,000 triangles and 150–175 draw calls per frame [17]. Put simply, this means that, considering performance, a very serious reduction was needed to be made to the number of polygons of the 3D models, and the number of materials, texture sizes and lighting had to be optimized.

On the other hand, the aspiration of the project was to provide a realistic representation of the four historic warships, enhanced with important information about the decisive Battle of Navarino. From the beginning of development, it was clear to us that a reduction of complexity in the 3D models or the quality of textures and materials was not an option: the project's depiction of the warships should be as accurate as possible.

Optimization methodologies adopted from the video game field were the answer to this problem.

- The high polygon count 3D models, derived from detailed drawings and NURBS surfaces (.3DM) were turned into a more common type of mesh, then remeshed and retopologized in order to reduce polygon count.
- In order to reduce draw calls, unwrapping and texturing was carried out in a manner that focused on the reusability of textures and materials on different meshes, resulting in a reduced number of materials. In addition, a tool was used to bake the lights in Unity3d.

- Occlusion culling was also implemented in order to reduce triangles and draw calls per frame.

In the following parts, the methodology employed for the development of the VR experience "The Ships of Navarino", will be presented and analyzed in detail.

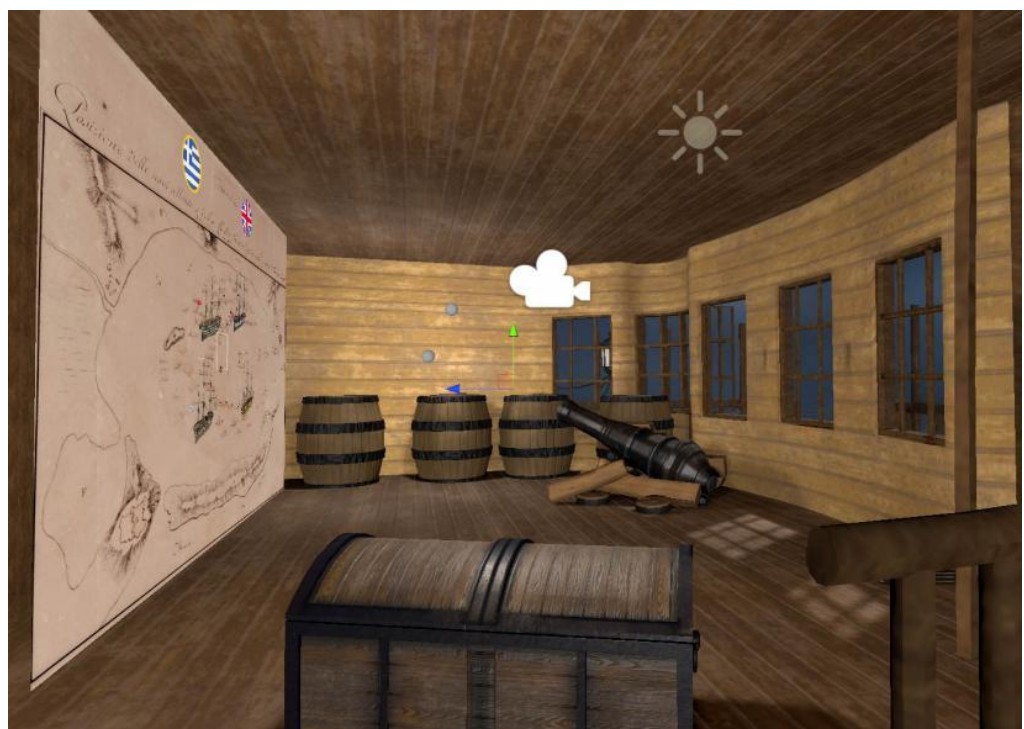

**Figure 1.** The introductory scene of the VR application "The Ships of Navarino". Screenshot from Unity3d.

## 2. Methodology

### 2.1. Ship Plans & 3D Model Creation in Rhino3D

The methodology is described in relation to only one of the four ships indicated above, the Russian Azov. For a better comprehension of the work in this project, the historic ship lines plan of Azov [18] was used for the creation of the 3D ship models in Rhino3D [19]. The historical research and documentation of the aforementioned blueprints was carried out by the NAVS project partners Institute of Mediterranean Studies, FORTH, and the Eugenides Foundation.

The geometry in Rhino3D is based on the NURBS [20,21] mathematical model, which aims to produce mathematically correct representations of curves and freeform surfaces [22]. The development of ship sections, as seen with red NURBS curves in Figure 2, is a basic method that is commonly used in ship design. These curves are smoothed out, in order to create a hull that is as smooth as possible.

In the next step, a subset of the faired curves was chosen to form a curve network that leads to a hull-surface. The surface was mirrored, and with the proper adjustments, the hull converted into a closed polysurface, which is the equivalent of a solid in Rhino3D. The hull, along with the keel, is shown in Figure 2 (green), with the selected sections (red) aligned in 2D and 3D space.

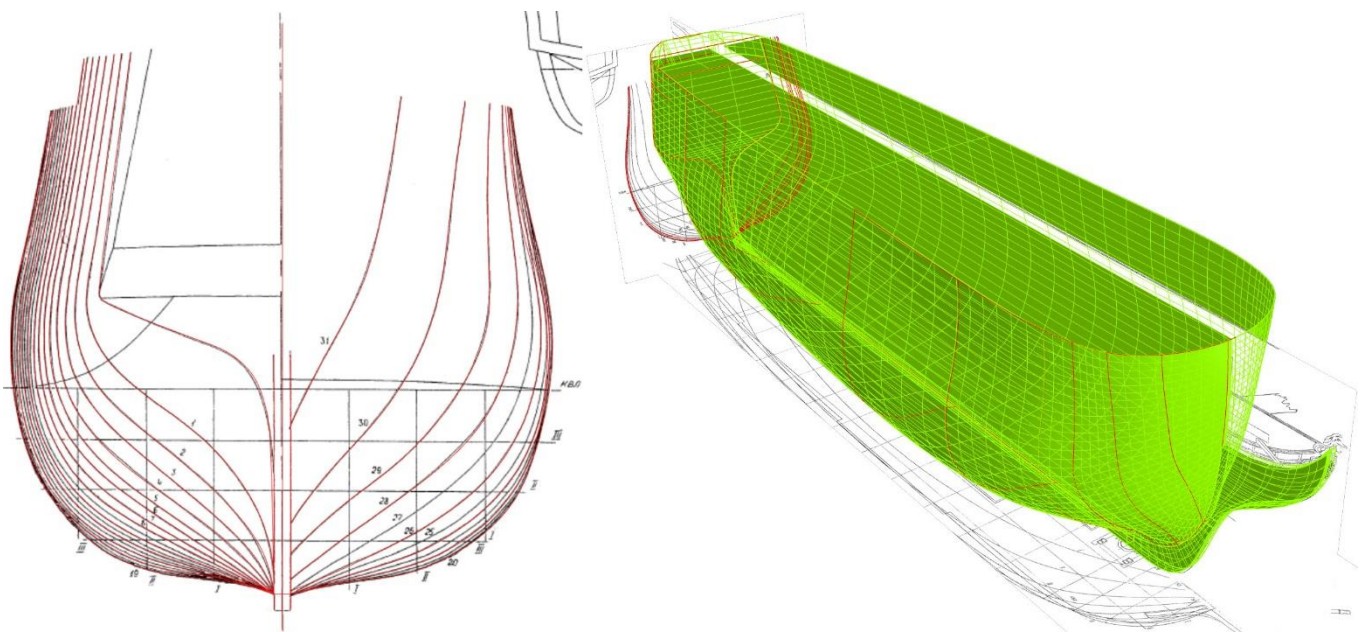

**Figure 2.** Drawings of the Azov Ship: left—lines plan; right—surfaces constructed in Rhino3D.

Proper cuts in the solid hull were made, as shown in Figure 3, to construct the deck flooring and the openings for the masts and cannons. Figure 3 also depicts the deck components of the Azov, as well as the longitudinal wooden strips that run outside the hull and follow the geometry of the outer surface.

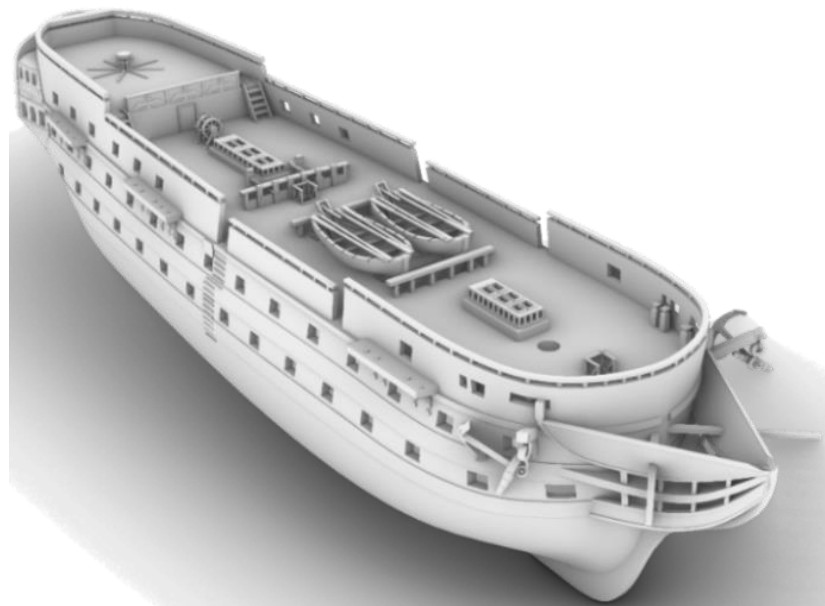

**Figure 3.** Hull, deck, and deck details (lifeboats, deck hatch covers, etc.). 3D models created in Rhino3D.

The full-rigged 3D model of Azov equipped with sails and masts is depicted In Figure 4. The sailing ropes are also shown in the same figure. Finally, with all the gunport windows open, the cannons can be seen (starboard side).

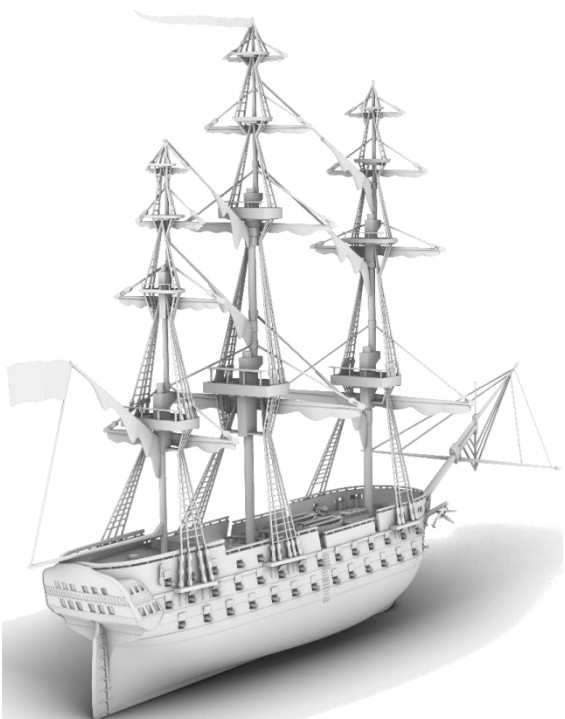

**Figure 4.** The fully rigged 3D model of Azov (from the stern).

*2.2. Polygon Mesh Optimization*

2.2.1. Conversion of a Rhino3D Model to a Polygon Mesh

As mentioned in the previous chapter, Rhino3D uses the NURBS mathematical model for the surfaces produced (Figure 5).

This method cannot be used by the other software utilized in this project, so it was necessary to convert all the surfaces from NURBS surfaces into another surface type, called a polygon mesh. Polygon meshes are collections of vertices, edges, and faces that define an object and can be used by any 3D-modeling and design software. There are many common file formats that can contain meshes, and for this project the .FBX file format [23] was chosen as it can contain many different objects separately, while also preserving the hierarchy of those objects.

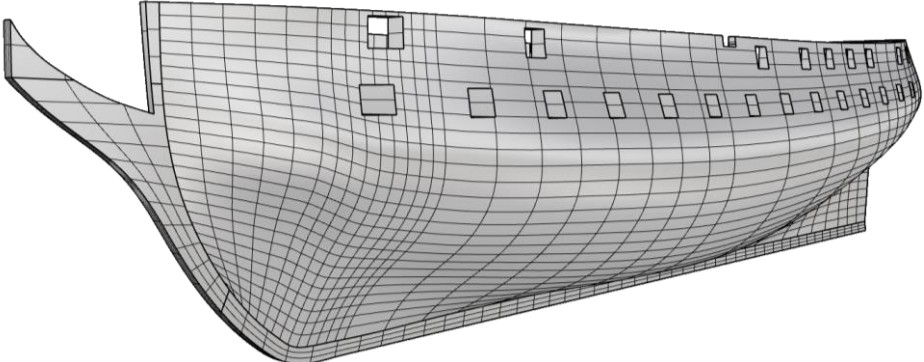

**Figure 5.** NURBS surfaces in Rhino3D.

In order to export a NURBS surface into a .FBX file containing all the polygonal meshes, a conversion from NURBS surface to polygonal mesh was deemed necessary. This conversion had to be as lean as possible, meaning as few polygons as possible while maintaining all the surface's information (geometry and curvature). Furthermore, in order to make future steps easier, with regard to unwrapping and texturing of the models (on

which more details are provided later), it was desirable to make all or most of the polygons quads, meaning that each polygon was defined by four edges and four vertices.

Rhino3D provides two ways for doing this conversion:

- Automatically during export, resulting in much undesired topology (many triangles, especially around holes in the mesh) making subsequent processes more difficult.
- Quad remeshing, which attempts to convert the NURBS surface into a polygon mesh containing only quads [24].

A quad-based topology means that a mesh is comprised mainly by quads. This is a generally preferable practice for 3D artists, for various reasons. A quad-based model is the most useful form for 3D modeling, as it ensures a clean topology and provides edge flow that can easily be adjusted. This is highly important when further editing is needed, for example when attempting to reduce polygon count, or when unwrapping is mandatory for texturing the mesh; as in this case. Secondly, subdividing a clean quad-based topology is a controlled and low-risk task, while creating edge loops in a primarily triangle-based mesh can be unexpected and cause shading issues that will be rendered into the game engines.

Whilst game engines such as Unity3d triangulate the 3D model during import, the triangulation of a quad is considered more predictable than that of an Ngon (a polygon with more than four edges), which may result in unwanted shadings and normal vector calculations.

Quad remeshing was chosen because it produced excellent quality polygon meshes that maintained all the original surface's information while being solely comprised of quads. The following figure (Figure 6) clearly shows, in contrast to the previous figure, that the mesh holes are completely defined by the quads around them, while the NURBS surface has no discernible way of defining the holes.

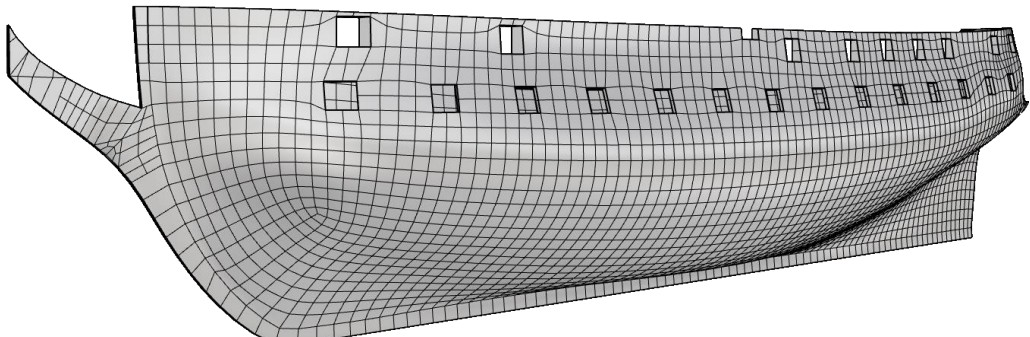

**Figure 6.** Result of quad-remeshing the NURBS surfaces of Figure 5.

The quad mesh does have some issues, especially in the bow area, but all those are refined during the retopology that takes place in the immediate step following this conversion. The quad remeshing tool could not be used on all objects of the models, so retopology was completed on those objects.

After the quad-remeshing process was complete, all the meshes created were exported into a .FBX file which was then used for retopology, unwrapping, and texturing of the models.

### 2.2.2. Retopology

Retopology is "the act of recreating an existing surface with more optimal geometry" and it is used to modify a high-resolution 3D model to its optimal form, to be utilized for animation or texturing [25]. The reason retopology was used in "The Ships of Navarino" was twofold. Firstly, since retopology is one of the most popular methodologies that 3D designers use to reduce polygon count, it was utilized for optimization purposes. Secondly, a low polygon model, with clean topology that provides edge flow, significantly reduces the UV mapping complexity, a process necessary for later texturing (Figures 7 and 8).

There are various options for retopologization:

- Manually, in a 3D-modeling software such as Blender. Given the built-in features of Blender, such as surface snapping and the shrink-wrap modifier, a designer can attempt to rebuild the high poly model polygon-by-polygon.
- Using specialized retopology Blender add-ons that attempt to reduce complexity and accelerate the retopologizing process.
- Using software outside Blender that offers more retopology tools.

Using some Blender retopology add-ons that allowed face snapping was the best option for this project.

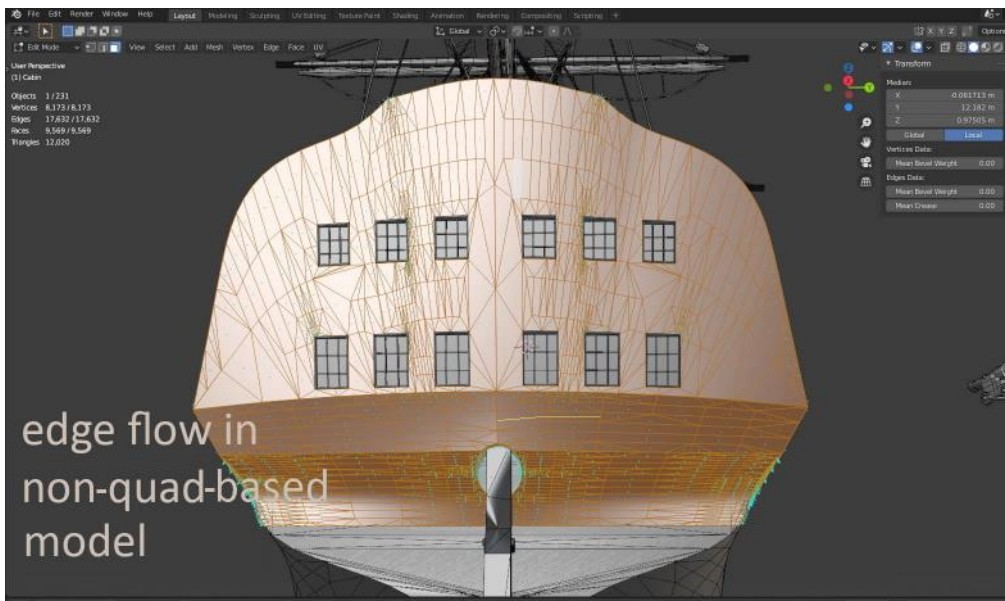

**Figure 7.** A non-quad-based model before retopology. Screenshot from Blender.

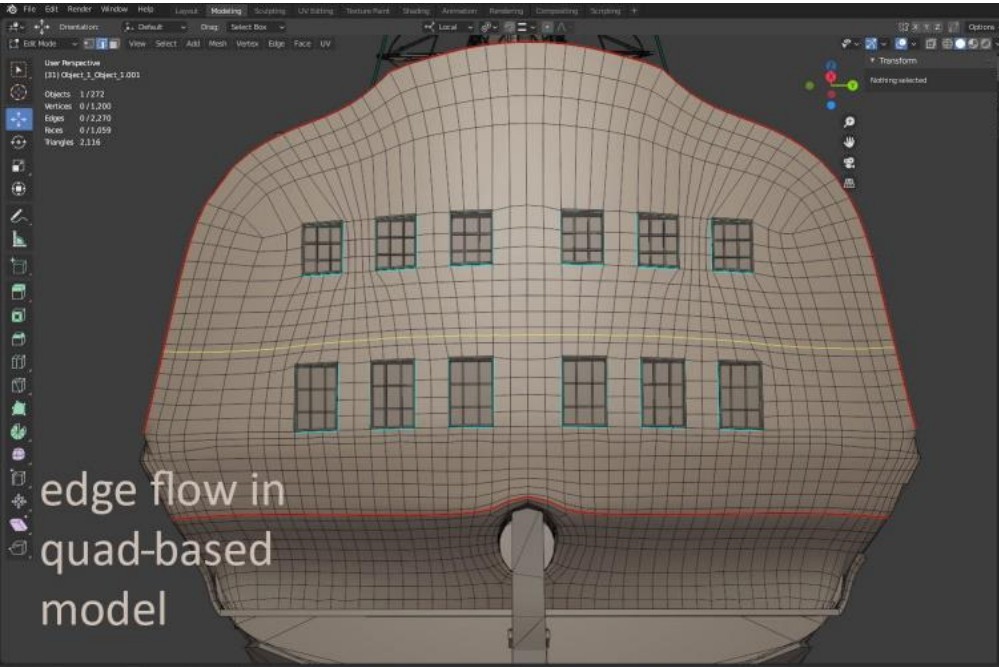

**Figure 8.** The same model after retopology with edge flow. Screenshot from Blender.

After remeshing, retopologizing, and refining the meshes, the most optimal form of the ship models was achieved. The final models counted significantly less polygons and were lighter and easier to manipulate.

Converting a high poly to a low poly benefits not only the modeling process but also the quality of the final model, because details of the high poly can be maintained without causing performance penalties (Figure 9).

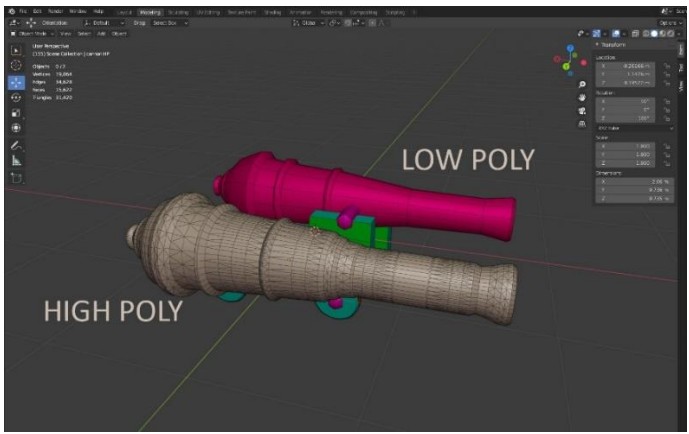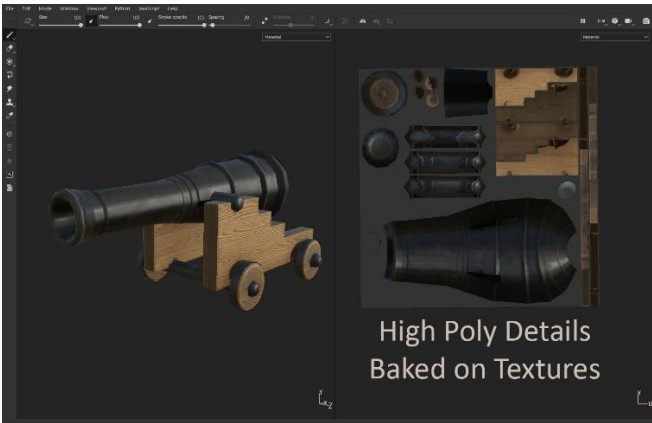

**Figure 9.** High poly to low poly workflow example. Screenshot from Blender and Adobe Substance 3D Painter.

By converting the high poly model into normal, occlusion, and other maps, details that demand a large number of polygons can be rendered into them and used as textures afterwards.

### 2.2.3. Texturing

3D Texturing refers to the process of creating detailed textures (maps) to be applied (mapped) onto the surface of a 3D model. In order for the texture to be applied correctly onto the surface, the model has to be prepared with a process called UV unwrapping.

### 2.2.4. UV Unwrapping

A UV map is the representation of the surface of a 3D model onto a 2D map that is later used for texturing the specific surface. The process of creating a UV map is called UV unwrapping. For 3D texturing it is necessary to unwrap all of an object's surfaces to the UV space. To further explain, the process works as follows: the unwrapped surface is projected onto a specific place in the UV coordinates, which enables the surface to display the texture information on the 3D model. An example of a UV-unwrapped texture is shown in Figure 10.

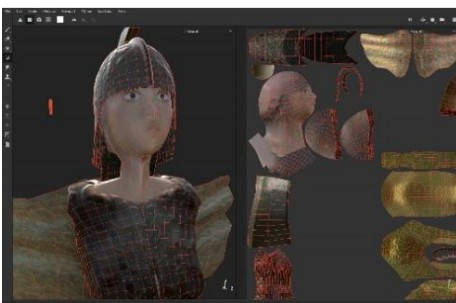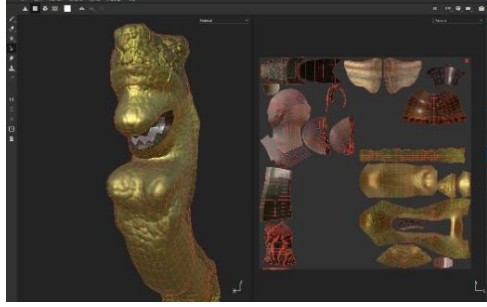

**Figure 10.** Example of UV unwrapping. Texture created with Adobe Substance 3D Painter.

The way the UV maps are organized is crucial for the texturing process. For example, if two or more surfaces are overlapping in the UV map, these surfaces will keep rendering

the same textured coordinates. In addition, the UV maps have to be distortion-free and maintain their proportion ratio, otherwise the texture appears stretched and disproportionate. UV unwrapping is also critical with respect to the texture quality. In fact, the larger the item on the UV space, the greater the density, meaning that there is more space available for painting details.

Another thing to take into account when unwrapping is the seams. A seam is the edge where the mesh surface is "cut" in order for the 3D model to be mapped onto the UV map. Seams can create visual discontinuation of the texture if they are misplaced, so when possible, they should be hidden in places not easily visible, or where the material changes.

All known 3D-modeling software include tools for UV unwrapping the 3D model, as this procedure is mandatory and texturing depends on that. "The Ships of Navarino", were unwrapped into Blender and their UV maps organized hierarchically by size, in a way that most space was held by the surfaces that were most likely to be seen closely by the user.

### 2.2.5. Applying Textures to a Polygon Mesh

When a 3D model is successfully unwrapped there are two options:

- Manually paint the textures onto the UV maps.
- Import the unwrapped 3D model into 3D texturing software and generate the painted textures.

3D texturing software, such as Substance 3D Painter, provide the necessary tools for painting high-quality and detailed maps (base color, normal, roughness, metallic, ambient occlusion, etc.), facilitating the creation of PBR materials. PBR stands for physical based rendering but it can also be known as physical based shading. It is a pipeline that simulates materials in a plausible way, aiming for realistic results.

Advanced masking and procedural texturing tools allow us to achieve textures that are much harder to create with a 2D program such as Gimp or Photoshop. Substance 3D Painter allows the creation of realistic materials using various sources and provides procedural tools to enhance the base materials with whatever is necessary for the most accurate resemblance. Finally, it bakes the textures to different maps according to the target engine, in this case the metallic channel of Unity3d's standard shader.

Reference images are some of the most important resources for the design of the textures and, when talking about a realistic representation of a historical event, are strictly tied with the quality of the result. For "The Ships of Navarino", there was access to a small archive of drawings and images resembling the warships, and to historical specialists that acted as counselors. On this basis the various materials of the ships were created. Given the fact that this project demanded careful spending on materials and texture sizes, a flexible organization of UV maps was used so that one texture would include more than one material and every material could be used in more than one mesh, and potentially in more than one ship.

To further elaborate on this, every texture was a blend of more than one material, depending on each ship's demand for different materials (Figure 11). Creating textures in this way also proved to be very useful in a later stage, when correcting a mesh's material was an easy procedure when could occur just by moving the UV tiles to different coordinates. However, due to their unique nature, specific meshes such as a ship's wheel (Figure 12), anchor, or figureheads, were treated differently, as they were solely hand-painted and in fact, in more detail, so that the materials could be used exclusively on them. Additionally, corrections on these objects were a more complex procedure as they could only be completed with Substance Painter.

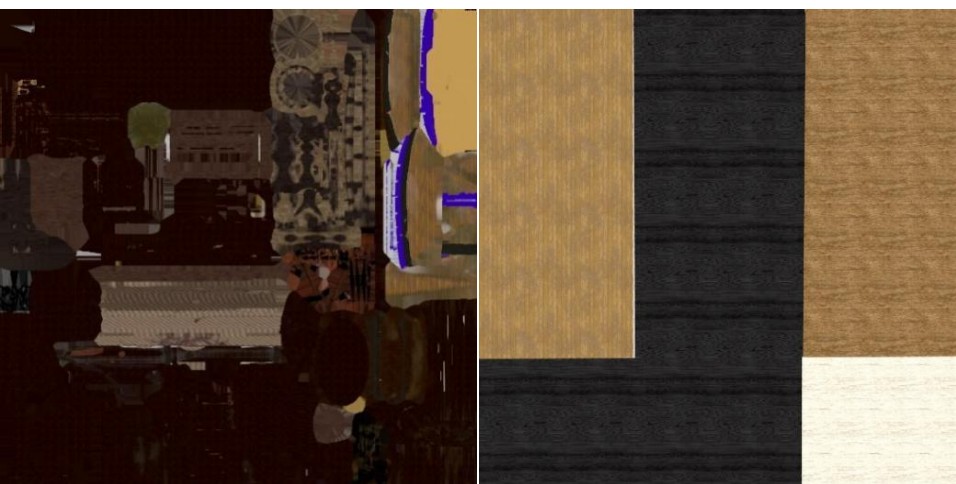

**Figure 11.** Maps created by Adobe Substance 3D Painter.

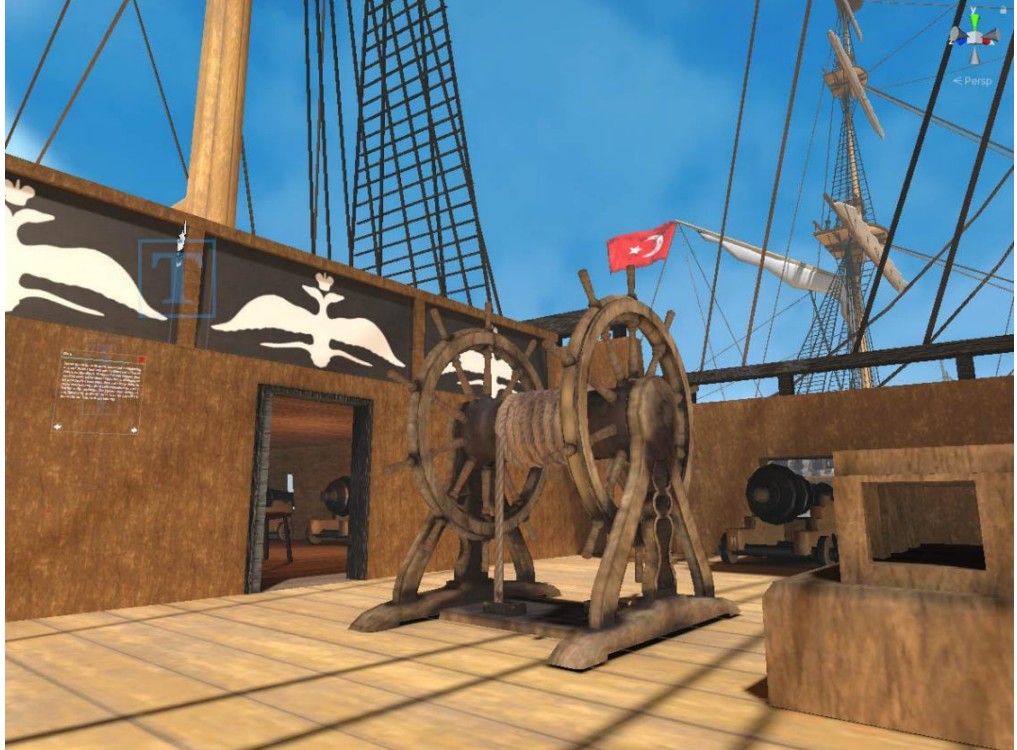

**Figure 12.** Azov's deck. Screenshot from the environment of the Unity3d Game Engine.

Following a pipeline that highlighted reusability, when possible, a significant reduction in draw calls was achieved, because inside Unity3d their number maintained a low reassuring performance whilst maintaining accurate representation of materiality on the ships (Figure 13).

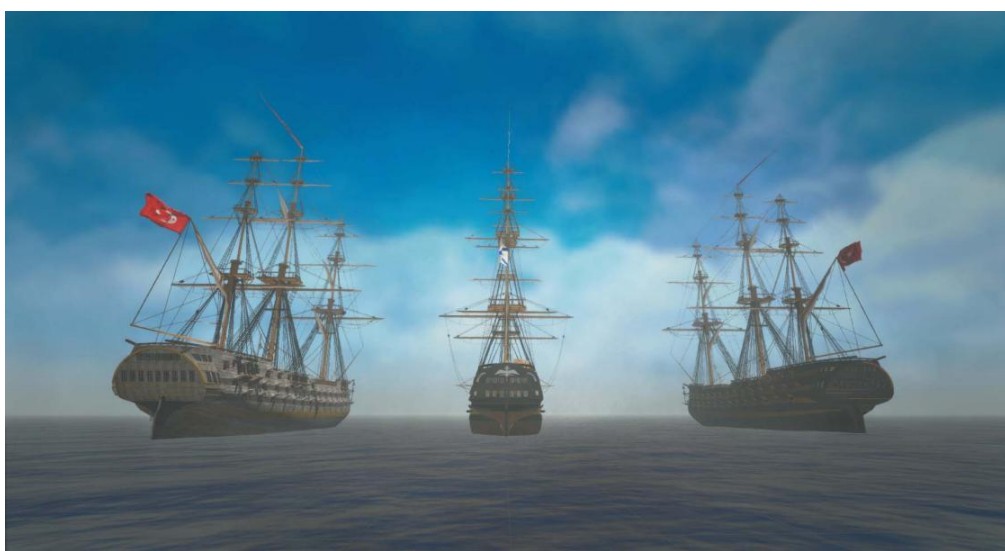

**Figure 13.** Ships from the VR application "Ships of Navarino". Screenshot from Unity3d.

### 2.3. Developing and Optimizing a VR Application in Unity3d

When developing a VR application, it is very important that optimization is taken very seriously. There are many steps to optimizing any application made in Unity3d, especially for a VR project. The goal of the project was to keep as much of the visual fidelity as possible while also obtaining acceptable performance, to avoid the implications of an inadequate or inconsistent performance (nausea, motion sickness). In this chapter, the optimization steps that were implemented are thoroughly discussed.

#### 2.3.1. Geometry Optimization to Reduce CPU Time

The first step is geometry and texture size optimization, both of which were already completed in the previous step. Since meshes require CPU time to be drawn, the fewer triangles they contain the better [26]. Textures use both the CPU and GPU, for loading and displaying, respectively. Lowering their size is important to avoid long loading times between scenes and potential VRAM limitations on the graphics card, which could lead to stuttering. Additionally, separating the different items into separate meshes is considered beneficial for the third step of the optimization process, occlusion culling.

#### 2.3.2. Choice of Lighting Method

The second step is using baked lighting instead of real-time lighting. Real-time lighting requires every triangle to be drawn multiple times in order to display the mesh, the texture that is applied on the mesh, and the shadows cast by the mesh or nearby meshes. Although this can produce more realistic lighting and shadows, it is very intensive on hardware and may produce lower than necessary performance, because VR is already intensive on the GPUs. On the other hand, baked lighting is precomputed and has a very small performance overhead, while also providing acceptable levels of lighting and shadow quality [27].

In Unity3d, lighting is separated into direct and indirect (Figure 14). Every ray cast from the light source that hits a mesh is considered direct lighting, and every surface that is lit from the bounces of those rays is considered indirect lighting [27].

Baked lighting was chosen in order to save as much performance as possible, given the fact that each frame is drawn twice for any VR application (once for each eye). For this method a Lightmapper was needed, which is a piece of software that uses light sources in a scene to precompute direct and indirect lighting and shadows cast from all sources, and then compresses the data into textures called lightmaps. These lightmaps were then overlaid on the textures applied to each mesh to give the illusion of lighting and shadows. Unity3d does offer a Lightmapper that is built into the engine, but it was decided to use a Unity Asset Store package named "Bakery GPU Lightmapper" because of its superior

speed and end result. Bakery's standard shader is equivalent to Unity3d's standard shader with a few extra features. For example, Unity3d's built-in Lightmapper does not include lightmap specular highlights while Bakey GPU Lightmapper does. This makes scenes using Bakery a lot more vibrant and realistic [28].

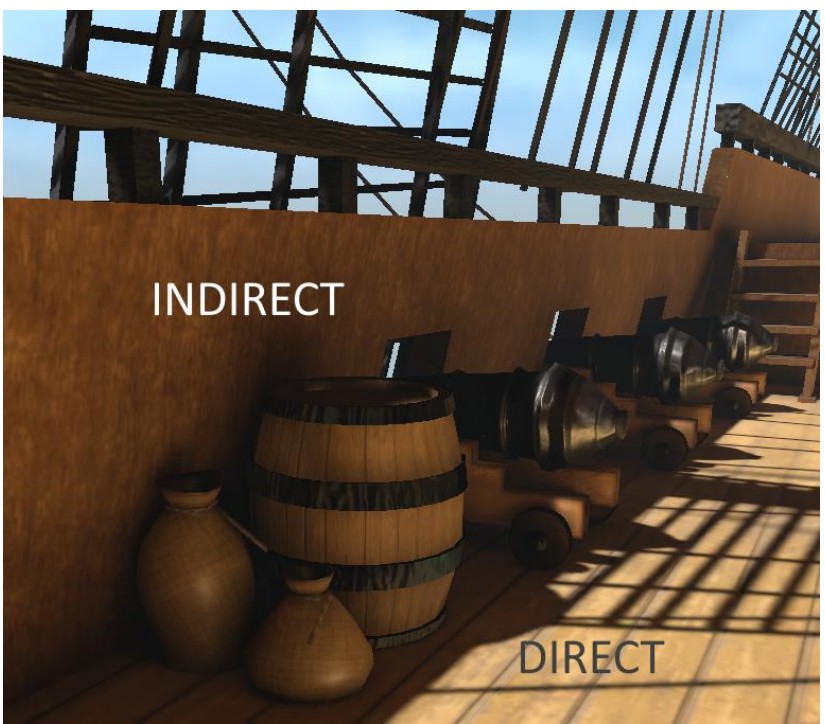

**Figure 14.** Direct and indirect lighting.

### 2.3.3. Occlusion Culling

The third step is occlusion culling. This is a process which prevents Unity3d from rendering objects that are outside of the frustum of the camera or hidden by other objects. This is completed as default in every frame, since the main camera performs culling calculations to determine which objects should not be drawn and can be supplemented by precomputed ("baked") occlusion for all static objects for better performance. This process saves both CPU and GPU time, because there is less geometry to be drawn per frame. The performance impact of this process is dependent on scene size, object count, and camera count [29] Only two cameras were used in the VR rig (one for each eye), aiming to balance the scene layout and object count so as to have as few objects as possible while occluding those necessary.

### 2.3.4. Achieving Visual Fidelity

A decent level of visual fidelity had to be maintained while keeping all optimization steps in mind. There are many factors that affect visual fidelity, but the two most important ones are shader workflow and post-processing.

#### Shader Workflow

Unity3d offers two shader workflows by default, the metallic workflow and the specular workflow [30]. The textures and texture maps that were created in the previous steps were designed with the metallic workflow in mind, because it was considered that it would offer a better representation of the actual material being in mind during the design. The specular workflow gave all materials a more "plastic" quality than actually desired, meaning some objects became too glossy when they should have been rough (e.g., the wood deck floor in Figure 15).

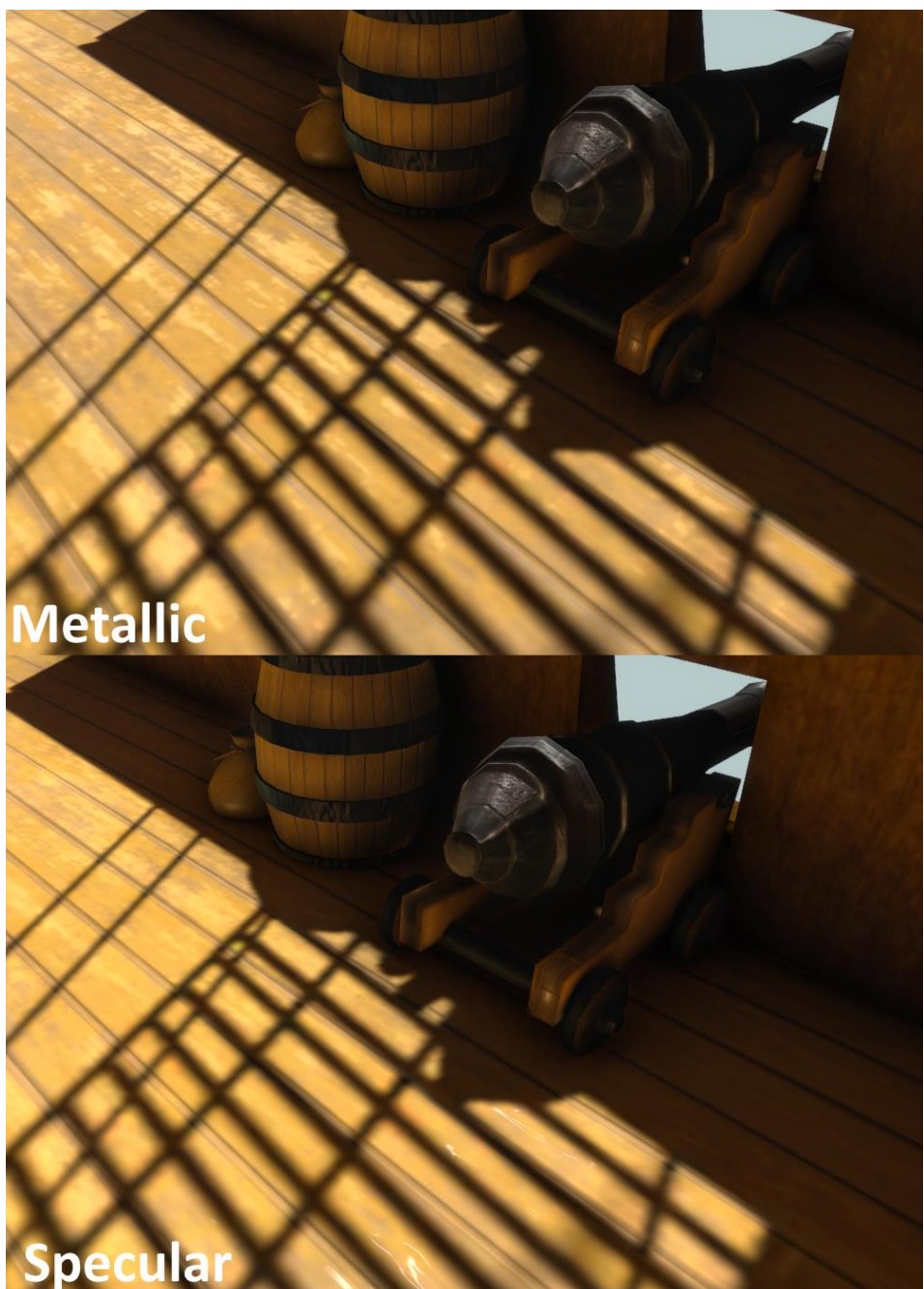

**Figure 15.** Top—metallic workflow shader; bottom—specular workflow shader.

Post-Processing

There is a variety of post-processing filters available for use in Unity3d. Only anti-aliasing was used to reduce the number of jagged edges, bloom to enhance the bright spots of the scene (mostly light reflections), ambient occlusion to emphasize the difference between directly and indirectly lit surfaces of the models, and color grading to give a better look to the models. Of note, is the slight orange-yellow tint added and the enhancement of the shadows, especially between the lifeboats (Figure 16).

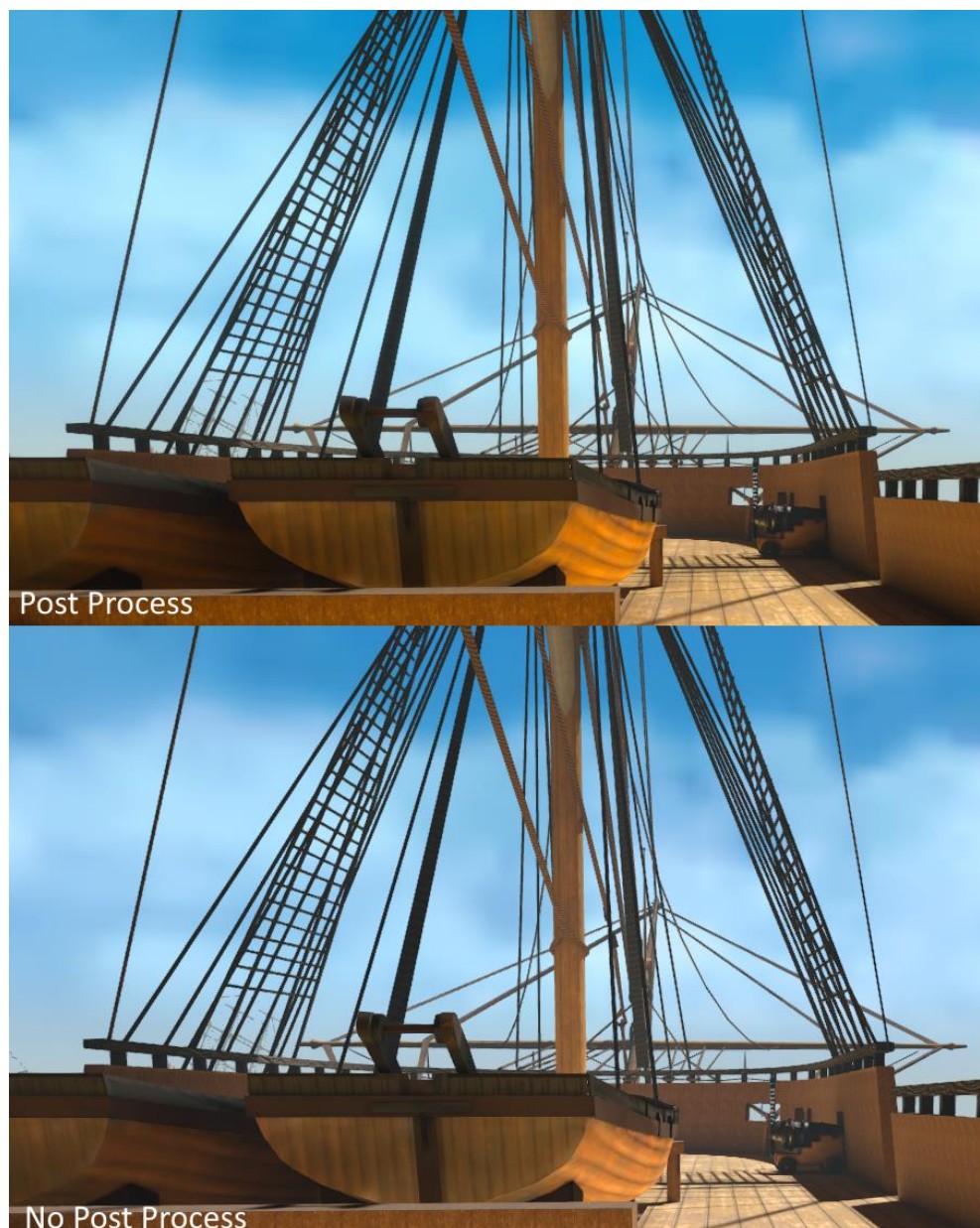

**Figure 16.** Top—scene with post-processing; bottom—scene without post-processing.

2.3.5. Adding Functionality to the Application

Unity3d uses the C# scripting language, so all the scripts are written in C# based on object-oriented and component-based architecture. The scripts were organized into three main categories: scene management, virtual reality implementation, and user interface.

Scene Management

Scene management is relevant to every three-dimensional object, audio, language management, and every function created to support this application besides VR scripts and UI scripts. The majority of the objects are static, due to the limited interaction with the user (museum limitation due to time constraints), so their management is restrained to updates of the player's position (ship, deck, cabin, masts).

There are some non-static objects, mainly the information points (tool-tips). Each information point indicates a position of interest, containing additional data (two pictures, 1. only info point, 2. opened info). All information points are buttons that display historical documents and pictures towards the player's field of vision. The scripts that were

developed for the information points are more complicated when compared to the rest of the scripts in this category. The aim was to develop scripts as generic as possible for the displaying images (multiple or none) in different sizes for each information point along with text (long or short). This was because information points had to be placed in various places on all ships (images for different aspects of information points: 1, only text; 2, text and one image; 3, text with two images side by side; 4, multiple text with multiple images).

As mentioned above, scripts were developed for audio and language management. We used background music and sound effects that change with the user's position and enhance the VR experience. Furthermore, scripts that allow the use of multiple display languages were created.

Virtual Reality Implementation

Virtual reality implementation scripts/functionalities pertain to the user's capability of moving in the three-dimensional world and the "bridging" between the Oculus device and the Unity3d engine. For the VR implementation it was decided to use the Unity asset "VR Interaction Framework" [31], with some adjustments to meet specific needs.

User Interface (UI)

The user interface (UI) script category concerns the interaction between the user and the 3D environment. The greater number of scripts that were developed for the UI involves buttons that are embedded in the 3D environment and buttons that are ingrained in the user (hands/controllers). Simple code sequences for the buttons were designed for two reasons: firstly, these button-scripts may be executed continuously, and complex code could affect the application's performance; secondly, the use of the buttons' functionality is limited to changing the player's position or to displaying objects (UI), so no complicated code chains are needed.

The VR Interaction Framework that was used contained models for virtual hands to give a better understanding of the user's position. Instructions were added, that can be toggled on or off, to each of the user's virtual hands in order to help the user identify the functions of each button and also to reduce the time that they would need to become accustomed to the VR controls. Controller models were also added to the virtual hands, to make it even easier for the users to orient their fingers on the controllers.

After setting up the scenes (one for each ship, plus one home scene), a map was created, on which there was one button for each ship. Each button has a different image and is easily discernible. Clicking one of the ship buttons brings up an information panel, which contains basic information about each ship, including the captain's name, crew number, ship type etc. The user can then click the "explore" button to transition to any selected ship. This map has been included in every scene and can be accessed at any time, provided the user is close enough to use the buttons. Each user starts on the home scene, so when the user takes the headset off it automatically transitions back to the home scene. Additionally, two buttons have been included in the map so that the user can change between Greek and English. The localization system is set up in a way that it is easy to add more languages, should that be necessary during the exhibit's lifespan.

Another way of transitioning between scenes was also created. Since the ships are large in size, it would be very inefficient for the user to be forced to return to the map in order to change scenes. Therefore, a portable UI panel was created. This is a simplified version of the map, as it only contains the images of the ships. The users can toggle this portable UI on or off with a button and transition between scenes from wherever they desire. This portable UI also allows the user to change freely between languages.

*2.4. Changes Based on User Feedback*

2.4.1. Fixes before Release

As with all applications, there were a few issues that occurred during the testing period. In order to fix all the bugs with the code and all visual issues faster, we consulted with people from the 3D application field (mostly friends and family).

The first issue was visual fidelity. At the start of development, much focus was placed on optimization, without giving enough weight to visual fidelity. However, with the techniques described in the previous chapter, such as baked lighting, this visual issue was fixed with no performance penalty.

The second issue was the way scene transitions were handled. At first, the only way was through the static map. The best suggestion was to create a portable UI that was simpler and smaller than the map (in order to avoid clipping with the surrounding geometry as much as possible), which was implemented as described in the previous chapter.

There were several issues with text displaying in the application outside of the Unity3d environment. These had to do with font quality and z-fighting, both of which were fixed.

Bakery has a mode for Lightmap baking called "SH" (short for spherical harmonics). This mode produces much more realistic results than the one used, but it did not function properly on the standalone application, so "full lighting" was determined to be the best choice. The reason behind the "SH" mode not functioning properly is unclear.

There was an issue with Oculus VR Interaction DLL, which rendered all of the VR Interaction Framework's functionality unusable. This was fixed with an update to that specific DLL. Additionally, a few issues with the Oculus Dash support in the application existed at first, but those were also fixed with an update to the same DLL.

2.4.2. Problems Identified after Release

The only major bug found after release was a bug referring to the portable UI described in the previous chapter. If the user was fast enough to click the "explore" button while the fade between scenes was still happening, the application would crash. Fortunately, this bug was quickly patched before the exhibit was opened to the public.

There were issues with the user's movement in a few places on the ships, which were caused by some misplaced colliders. This was quickly fixed, otherwise it made some areas of the ship inaccessible.

There were some minor changes on the flag textures of the ships, because the initial flags did not match the actual flags used by the navies in the 1800s.

## 3. The Ships of Navarino

*3.1. 3D Models of Four Historic Ships*

In total, four historic ships were created: the French frigate "Armide", the British "Asia", the Russian "Azov", and the Ottoman "Kuh-I-Revan", which can be seen in the following Figures 17–20.

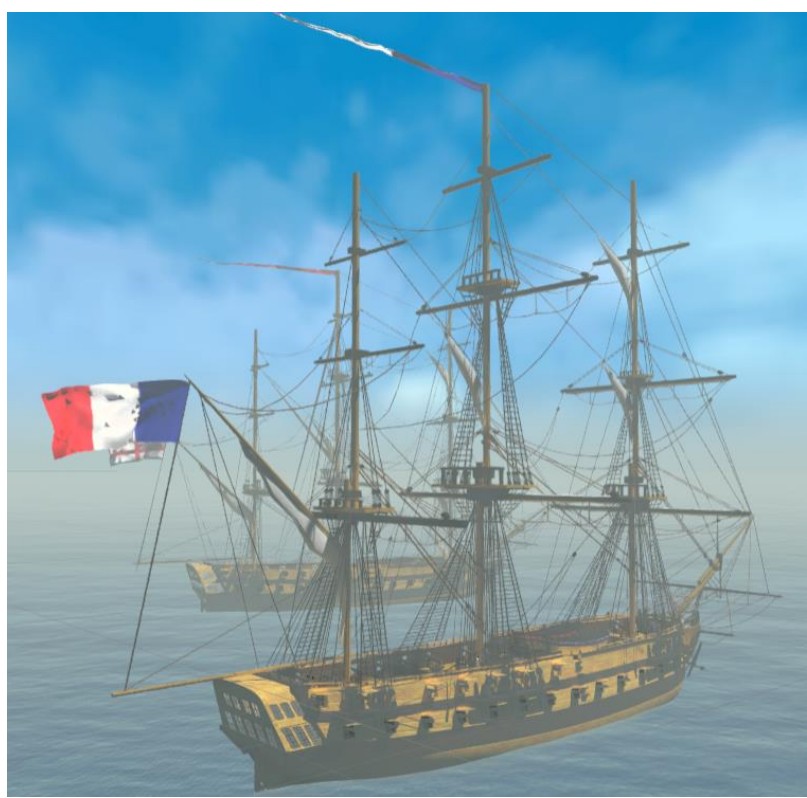

**Figure 17.** The French frigate "Armide". Screenshot from Unity3d.

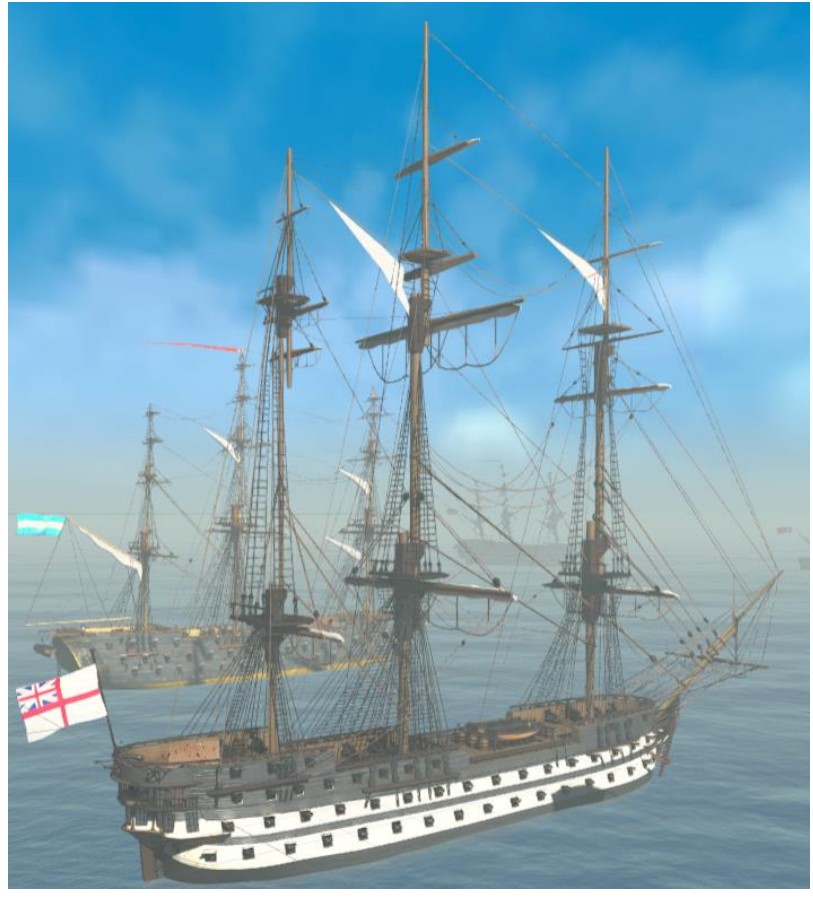

**Figure 18.** The British "Asia". Screenshot from Unity3d.

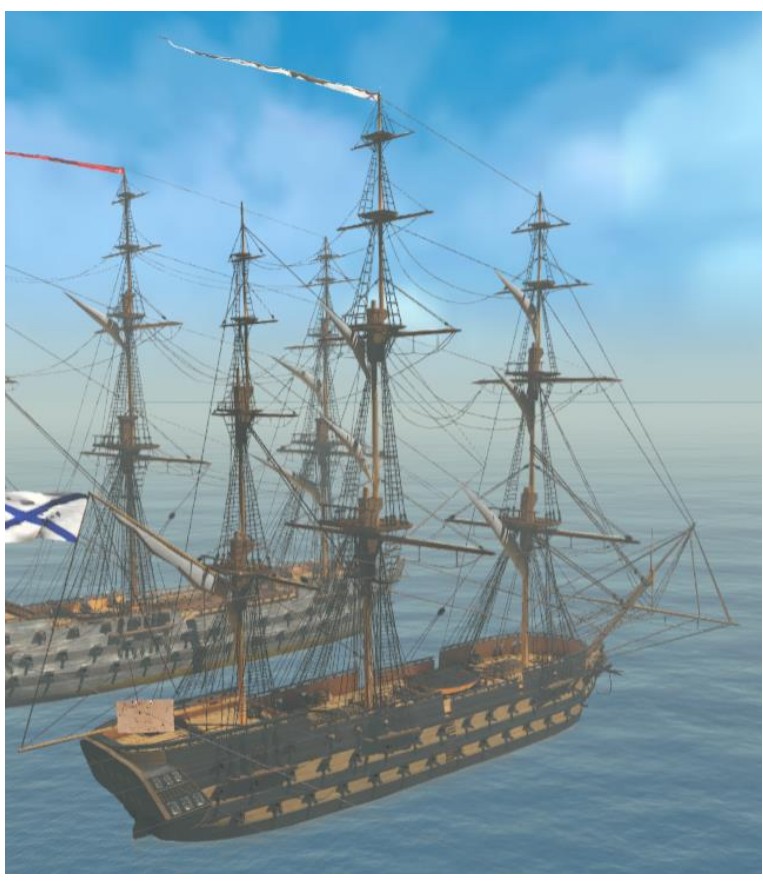

**Figure 19.** The Russian "Azov". Screenshot from Unity3d.

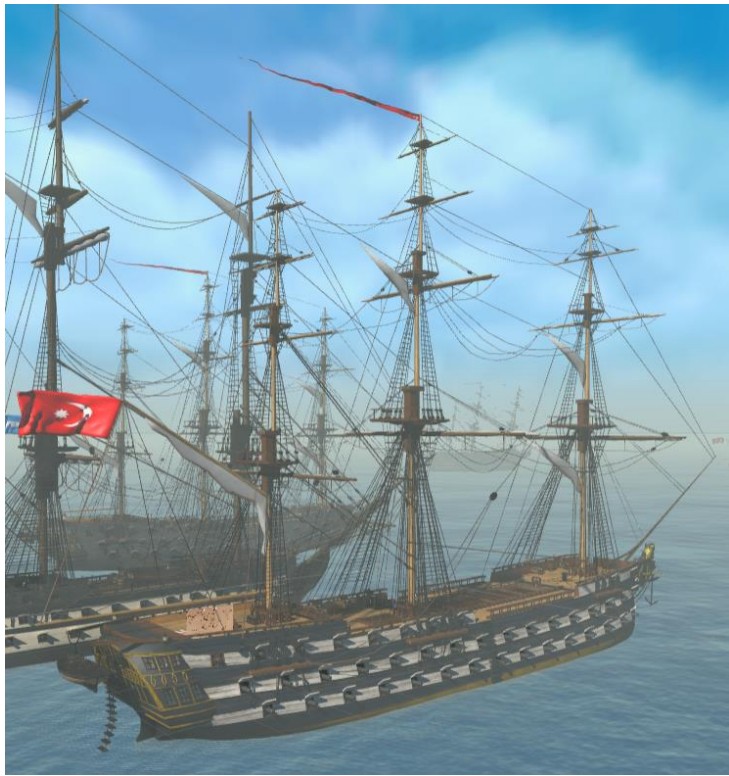

**Figure 20.** The Ottoman "Kuh-I-Revan". Screenshot from Unity3d.

The user can explore the main decks, the cabins, and the masts of these ships in the VR application, and read historic information about them during the naval battle of Navarino.

### 3.2. VR Application

The latest version of the VR application created is available to download from the NAVS project website [1]. There is also a preview of the application on the same site.

### 3.3. 3D Printing Process

The four ship models were printed on an SLA 3D printer called iSLA500 [32].

Stereolithography (SLA) [33] is a type of 3D-printing technology that operates by using a UV laser on a vat of photopolymer resin. This technology uses the UV laser and photopolymerization to turn photosensitive liquid resin into 3D solid polymers [34], layer by layer, creating the desired models.

Several review articles [35–37] have appeared recently addressing SLA 3D printing.

The 3D printing volume is 300 mm × 300 mm × 300 mm, with a 0.1 mm dimensional printing tolerance. The models are all scaled at 1:125, which is the same for all four Navarino ships.

After finishing the design in Rhino3D, the 3D-printing process began (see Section 2.1). A .3DM [38] file was converted to a .STL [39] file format. This file format is commonly used for 3D printing and is supported by a variety of software packages. The .STL files define only the surface of the full model geometry, without color or texture representation, through a triangulation procedure [40].

The .STL file was then loaded into the slicer software of the printer [41] to prepare the selected model for the 3D printer by generating a corresponding G-code [42]. The G-code instructions were sent to the 3D printer extruder, specifying all linear movements as well as commands for controlling the temperature of the extruder and/or positioning support objects on the 3D model (Figure 21).

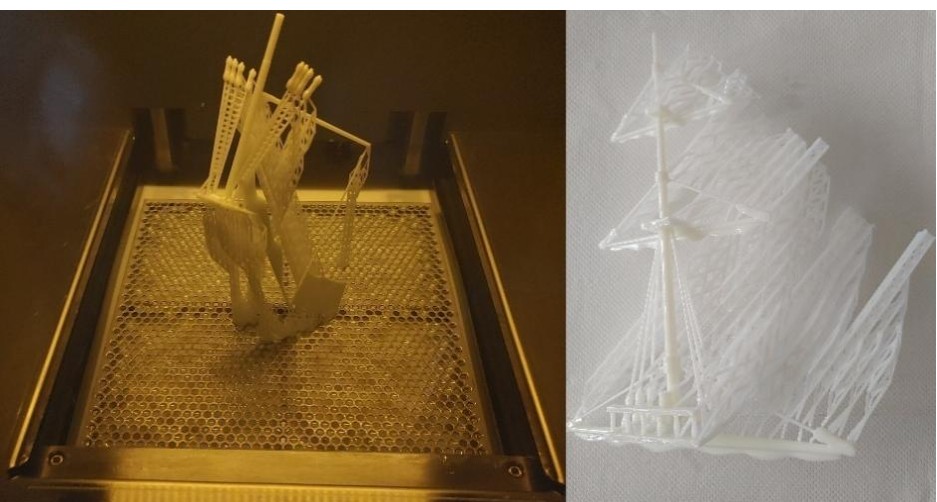

**Figure 21.** Left—printing process; right—3D-printed product.

Due to the printing volume constraint of the specific printer, the ship models were printed into parts, in order to generate as large a model as feasible, as seen in Figure 22.

In addition, the printed object was brushed and rinsed in isopropyl alcohol (IPA) to remove any remaining uncured resin. The printed parts were then placed in a UV curing chamber [43] (chamber model: PCA600). Their hardness and stability are improved by this post-curing process.

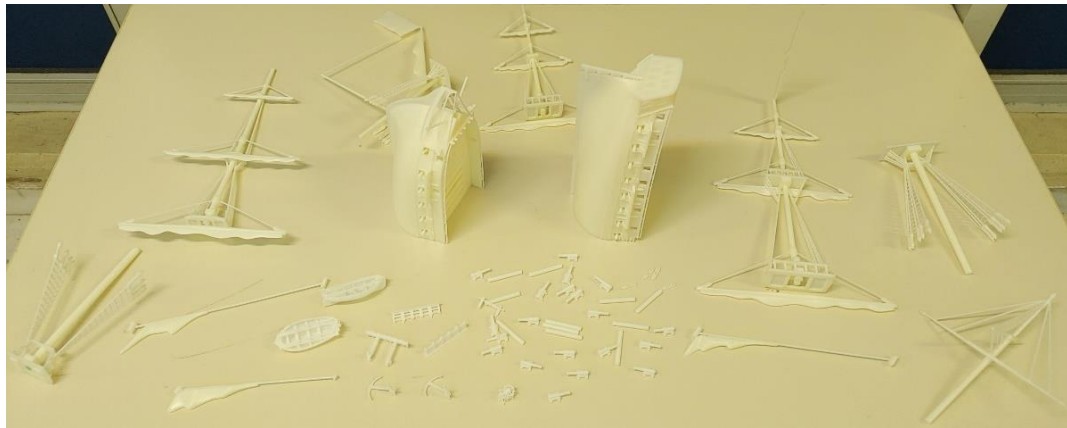

**Figure 22.** Finished products after alcohol rinse and UV cure.

Finally, the aforementioned supports (see Figures 21 and 22) were removed from the parts of the ship model, and the SLA parts were assembled and painted by an artist. The four final ship models are shown in the following Figures 23–26.

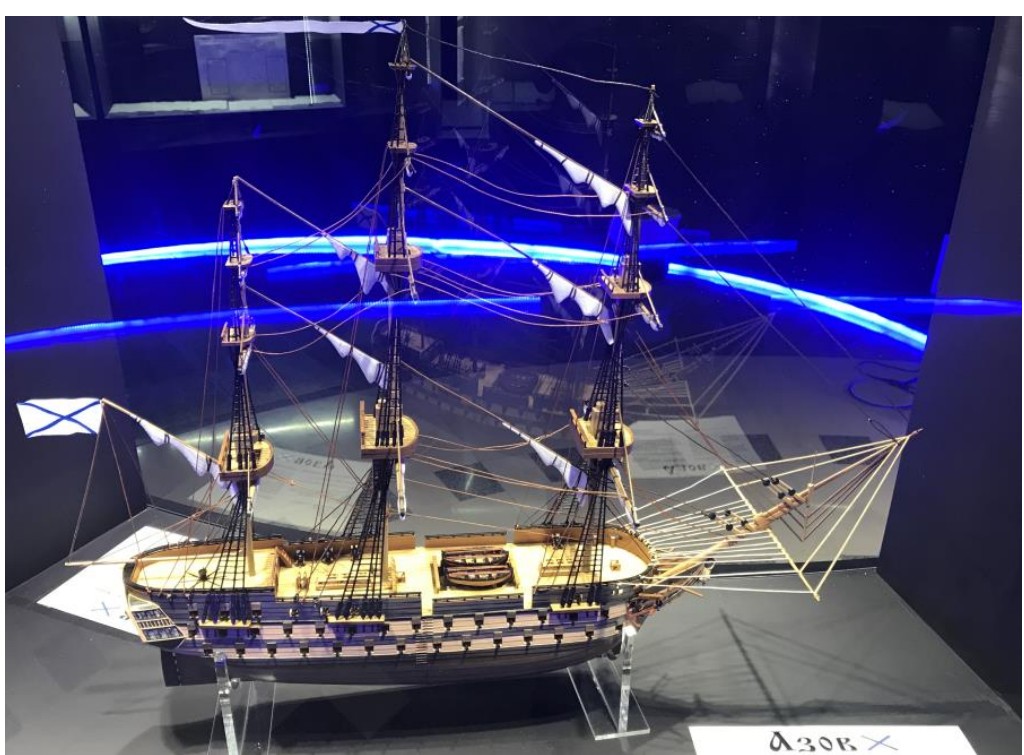

**Figure 23.** Azov 3D printed model after painting process.

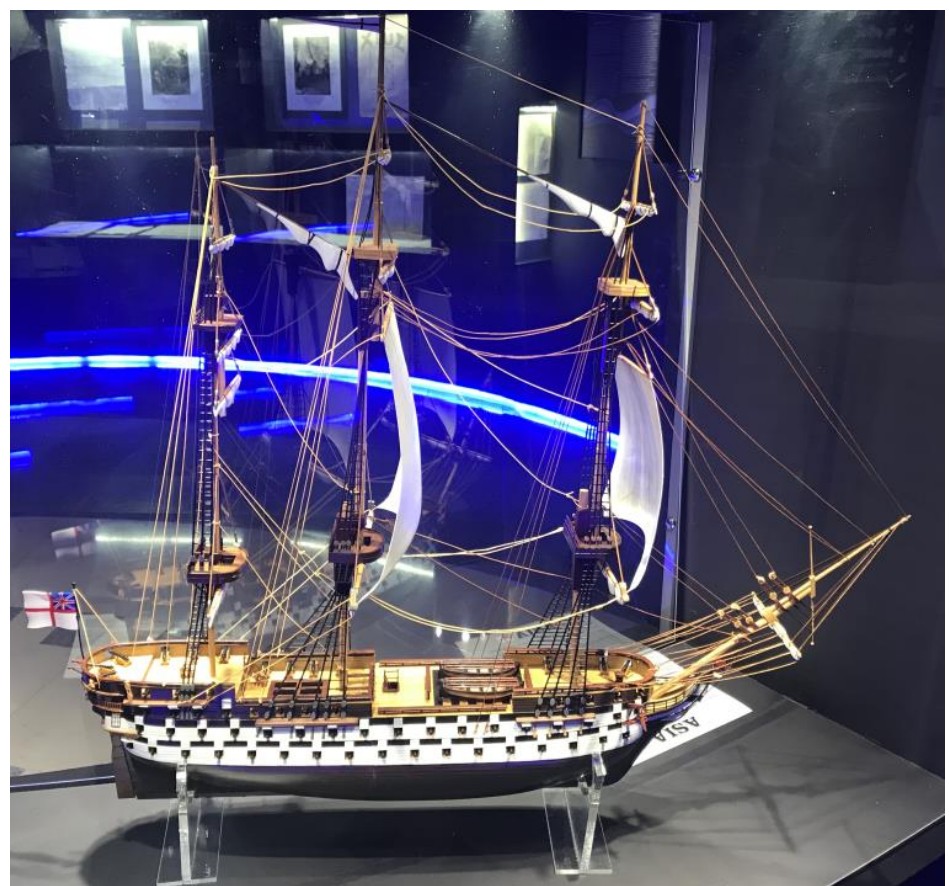

**Figure 24.** Asia 3D-printed model after painting process.

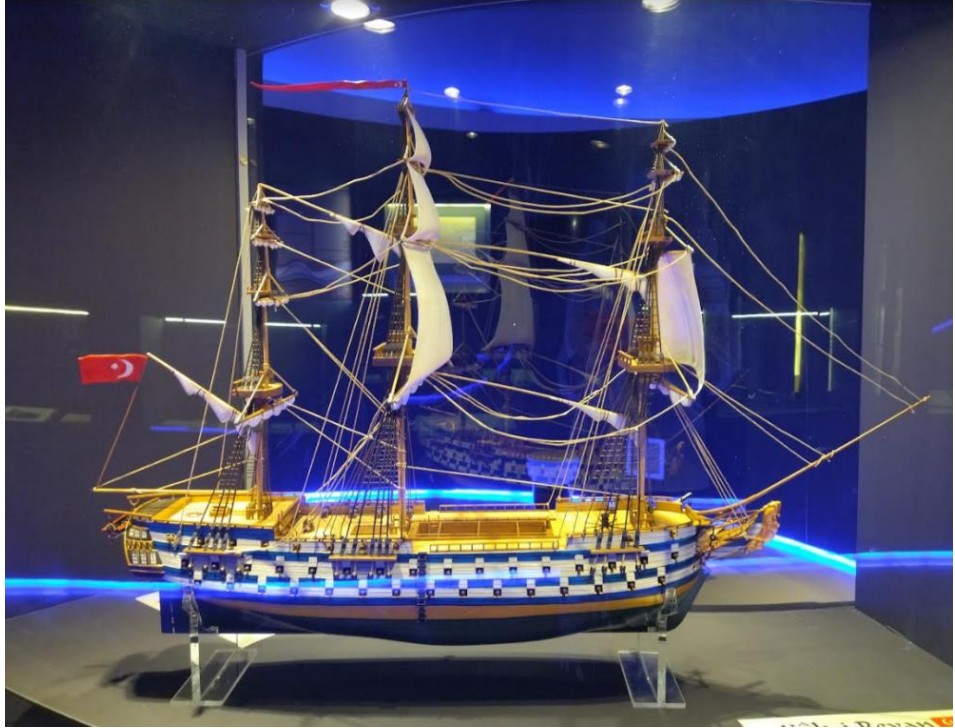

**Figure 25.** Kuh I Revan 3D-printed model after painting process.

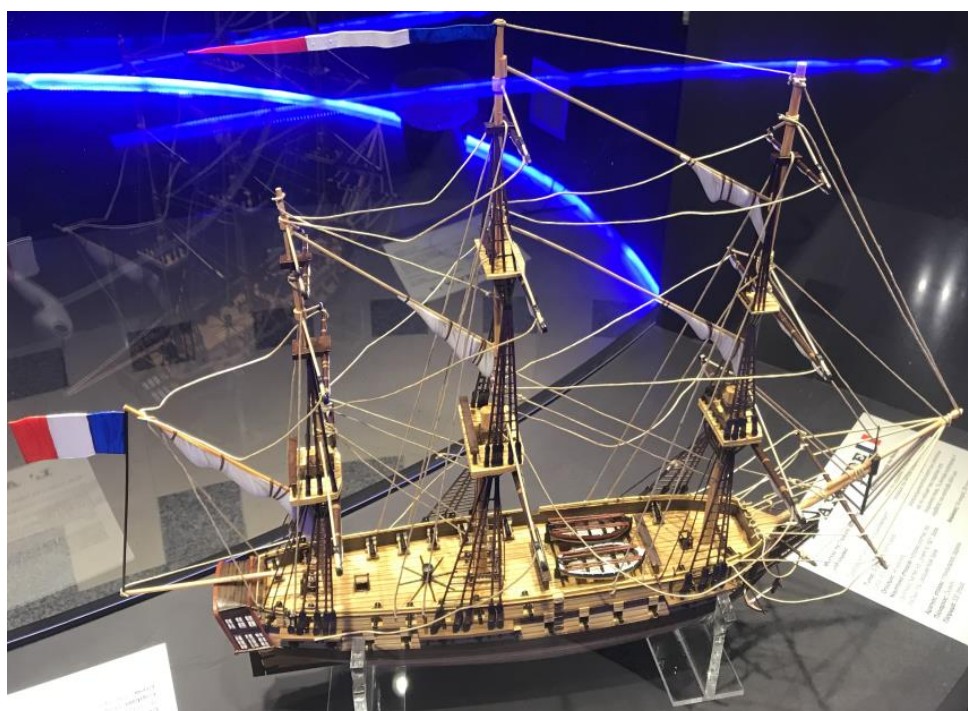

**Figure 26.** Armide 3D-printed model after painting process.

## 4. Discussion

This paper describes how VR technology and 3D digital modeling were used to recreate the historic Battle of Navarino, contributing to the preservation and wider dissemination of historical information to a global audience. This work was carried out as part of the NAVS project, and was launched in a special section at the exhibition of the Eugenides Foundation in Athens, Greece.

The optimization process for the VR application is explained in detail, including geometry optimization to save CPU time, baked lighting selection to save as much performance as feasible, occlusion culling to save both CPU and GPU time, ways to achieve adequate visual fidelity, and application functionality. The optimization approaches adopted, targeting the VR platforms, resulted in 3D models that could also be hosted online, enhancing the distribution of the project. Furthermore, accurate models of the four ships involved in the historic battle were digitally recreated, 3D-printed, and painted to be presented in the museum exhibition.

**Author Contributions:** Conceptualization, O.L., S.M., A.A., A.G. and C.P.; Data curation, G.R.; Funding acquisition, A.G. and C.P; Investigation, O.L., S.M., S.P. and G.R.; Methodology, O.L., S.M., G.P. and S.P.; Project administration, O.L., A.A., A.G. and C.P.; Resources, O.L.,S.M., A.A., G.P., A.G. and C.P.; Software, O.L., G.P. and A.G.; Supervision, O.L. and S.P.; Validation, S.M. and G.P.; Visualization, O.L., S.M., A.A. and G.P.; Writing–original draft, O.L., S.M., A.A. and G.P.; Writing–review & editing, O.L., S.M., A.A., A.G., C.P., S.P. and G.R. All authors have read and agreed to the published version of the manuscript.

**Funding:** The NAVS Project—promotion, documentation, and technical support of "The Greek shipbuilding legacy—the Battles of Navarino and Salamis" launched in June 2020, co-financed by the European Regional Development Fund of the European Union and Greek national funds through the Operational Program Competitiveness, Entrepreneurship and Innovation, under the call RESEARCH–CREATE–INNOVATE (project code: T1EDK-05103).

**Institutional Review Board Statement:** Not applicable.

**Informed Consent Statement:** Not applicable.

**Acknowledgments:** The authors would like to thank all those who participated in the project. In particular, Green Maritime Technology (GMT) who contributed to the design of the 3D ship models and artist Gigourtakis Yannis who painted the 3D-printed ship models. Additionally, the team would like to thank E. Nomikou and C. Troumpetari from the Eugenides Foundation for commenting on drafts of this paper, as well as associate researcher A. Pagonas for his support on the development of the VR application.

**Conflicts of Interest:** The authors declare no conflict of interest.

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
