# Peer review of "Development of the Virtual Reality Application: “The Ships of Navarino”"

_applsci, doi:10.3390/app12073541_

Round 1

Reviewer 1 Report

I want to congratulate the authors for the extensive work here presented. I have some things to say that perhaps could make this work a little better.

Firstly, about the writing, I would not use the first person in plural, avoiding the word “we” during the text. It sounds not technical including this kind of verbal time. Furthermore, I suggest the authors to review the way in which a software is called in the text. I have seen “blender” instead of “Blender”, while “Rhino” is written with capital letter. Also, I have seen differences in the way in which Figures are refered along the text, sometimes they appear in bold, some other times they don’t. To conclude with the writing, I would avoid the saxon genitive, being not formal language. The space between paragraphs I think is not allowed in the MDPI template.

To continue, I have to say that I miss a section called results. Perhaps, I would include the 3D print process (by the way, revise the way to refer to 3D-printing, I have read it “3d” and “3D”, please keep homogeneus writing along the entire text, if you once refer to a process, software o word in a way, you have to keep it all the time. For the reader is confusing this kind of changes), in the Material and Methods section. In results you can show the final products, and evaluate the quality of them (the VR production and the 3d printed model). It could be interest also to evaluate how people interact with these products, to have a feedback about the quality. Or maybe it can be included the time of processing and the time of printing that have been necessary. In conclusion, I can differentiate what the results are in this work.

Line 368 there is a double space between two words.

Figure 16, perhaps is better to invert the order, first the non process. It could be clarified the difference between them because they seem really similar.

Lines 533 and 534, it is necessary to have a space between the different figures references.

Thank you for your work, I hope I had made constructive suggestions.

Reviewer 2 Report

The team has given a good presentation of an extremely complex task: To create lively images, models and "films" of 5 different historic ships. On the base of two publications (Souvenirs de Marine - Collection Plans ou Desins ..., Paris 1882-1908., Nowavcky/Lefèvre, Creating Shapes and Naval Architecture - A Cross-Disciplinary Comparison Brill, Leiden/Boston 2009) the reviewer believes to have some insights in early ship design (16th-19th C). To him it seems that most ships of wood were made by elements crossing each other in a distant level, often in a vertical plane and in more or less horizontal curved way. The CAD-representions do not show much of this structural part of designs. The reason is probably that the authors tended to work with morph-like shapes, giving the exterior shape of the ships as a continuos body. One reason may be to reduce the digital information (Gigabyte) necessary for a detailed image, frequently discussed in variant surface representations by the authors. But for a user, a museum visitor or expert, this may cause a sense of loss of reality of the image. Especially in such cases where interiors or railings or gun holes are shown, missing parallel lines and a sense of structure (German: Gliederung) of similar elements (Fig. 1,12,14).  A better example is the deck planks in Fig. 15, which snow nerve directions which are natural. The reviewer dislikes in many cases images of wood which have nerve in transvers direction. He himselve opts than for surface applications which are neutral (non- directional) like Sand or a Lacque, or Metal. Better a more abstract image than one with unnatural appeal. To save digital information one can reduce lines: so one could save a rope not as a circualar tube  but as a hectogonal cylinder. At least we do this in ArrchCAD24.    

For a good result of depicting a ship, the water line is important. Of course the water line changes during a turn, and because of more or less loading. Fig 19 the Russian ship "Azov" looks rather unnatural, the sense of expected weight is less visible. Please   see web images of paintings. An expected 1 m or 1.5 m distance from the water line seems more realistic.

Please note: In this extremely interdisciplinary paper of a vast team I as a reviewer detect many aspects of which I have only little theoretical background. I even hesitated and did think of stopping the review on ground of "too little knowledge". On the other side I have worked as a trained architect with CAD application since 1984, mostly in close relation to students and CAD experts. So I know the general problems of a live-like image with CAD renderings. And I have close knowledge of Complex Geometry, mostly linked to Construction History. 

An writing error is the different representation of Three dimensions: Please choose either 3d or 3D. 

Reviewer 3 Report

Nothing to say. The topic is very interesting and the research is clear and well described. The final results are very good. The references are appropriated. 
